# Biasing Influence of ‘Mental Shortcuts’ on Diagnostic Decision-Making: Radiologists Can Overlook Breast Cancer in Mammograms When Prior Diagnostic Information Is Available

**DOI:** 10.3390/diagnostics12010105

**Published:** 2022-01-04

**Authors:** Fallon Branch, Isabella Santana, Jay Hegdé

**Affiliations:** 1Department of Neuroscience and Regenerative Medicine, Medical College of Georgia, Augusta University, Augusta, GA 30912, USA; fabranch@augusta.edu; 2College of Science and Mathematics, Augusta University, Augusta, GA 30912, USA; isantana@augusta.edu; 3Department of Ophthalmology, Medical College of Georgia, Augusta University, Augusta, GA 30912, USA; 4James and Jean Culver Vision Discovery Institute, Augusta University, Augusta, GA 30912, USA; 5The Graduate School, Augusta University, Augusta, GA 30912, USA

**Keywords:** anchoring and adjustment, bias, cognitive rules of thumb, diagnostic radiology, heuristics, judgement, medical error, screening mammography

## Abstract

When making decisions under uncertainty, people in all walks of life, including highly trained medical professionals, tend to resort to using ‘mental shortcuts’, or heuristics. Anchoring-and-adjustment (AAA) is a well-known heuristic in which subjects reach a judgment by starting from an initial internal judgment (‘anchored position’) based on available external information (‘anchoring information’) and adjusting it until they are satisfied. We studied the effects of the AAA heuristic during diagnostic decision-making in mammography. We provided practicing radiologists (*N* = 27 across two studies) a random number that we told them was the estimate of a previous radiologist of the probability that a mammogram they were about to see was positive for breast cancer. We then showed them the actual mammogram. We found that the radiologists’ own estimates of cancer in the mammogram reflected the random information they were provided and ignored the actual evidence in the mammogram. However, when the heuristic information was not provided, the same radiologists detected breast cancer in the same set of mammograms highly accurately, indicating that the effect was solely attributable to the availability of heuristic information. Thus, the effects of the AAA heuristic can sometimes be so strong as to override the actual clinical evidence in diagnostic tasks.

## 1. Introduction

One of the most impactful insights from several decades of research in human judgement and decision-making is that human beings do not always act as rational decision-makers but often resort to using heuristics (or mental shortcuts) when making decisions under uncertainty or under time pressure [1,2,3,4,5,6]. While using heuristics does have its advantages [3,5], the downside is that judgments (or, synonymously, estimates) based on heuristics can result in systematic errors, or biases [4]. Importantly, the use of heuristics is a fundamental faculty of the human mind and is known to occur in many diverse areas of decision-making and in novice subjects, as well as highly trained experts [7], including research scientists [2,3,4] and medical professionals [8,9].

Previous studies have identified many different types of heuristics that human subjects use [2,3,4,5,6,8,9]. Among the best-known heuristics are representativeness, anchoring and adjustment (AAA), and availability [2,3,4].

Now there is increasing recognition that many of these heuristics play a crucial role in clinical decision-making (for recent reviews, see References [10,11,12]). Indeed, previous studies have characterized the effects of some of the heuristics in clinical settings [9,12,13]. However, the role of the AAA heuristic in clinical decision-making remains poorly understood. The present pilot study aimed to help fill this gap by characterizing whether and to what extent the AAA heuristic affects diagnostic decision-making, using mammographic decision-making by practicing radiologists as an illustrative example (see Discussion for additional caveats).

In classical demonstrations of the AAA heuristic (and of heuristic judgement in general), the investigators verbally provided the subjects with a hypothetical problem scenario, or vignette, requiring a judgment under uncertainty and asked the subjects to make a judgement (or, in decision theoretical terms, an estimation). Studies have shown that subjects tend to begin with a starting (or ‘anchored’) judgment influenced by the ‘anchoring information’ provided by the vignette and adjust their judgement until they are satisfied [2,3,4]. Their final judgment usually reflects an error, or bias. This is because when the initial judgment is biased, subsequent adjustments rarely succeed in precisely offsetting the initial bias [14,15].

However, diagnostic decision-making (indeed, most real-world decision-making; see Discussion) differs from the aforementioned AAA study paradigm in two important ways. First, diagnostic decisions must necessarily be based on empirical, “bottom-up” evidence, as opposed to purely cognitive, “top-down” information (for an overview, see References [16,17]). Second, diagnostic decisions must be made on a case-by-case basis, so that the problem is essentially unique each time [13,17]. For instance, a radiologist deciding whether a given breast is positive for breast cancer must ultimately base the decision on the actual mammogram/s of the individual patient. Whether or to what extent this class of decisions is subject to the biasing effects of anchored judgements remains almost entirely unclear. The present pilot study directly addresses both these issues.

## 2. Materials and Methods

Subjects. All procedures used in this study were duly reviewed and approved in advance by the Augusta University Institutional Review Board (IRB), and all methods were carried out in accordance with relevant guidelines and regulations. All subjects were adult volunteers who provided written informed consent prior to participating in the study. All subjects had normal or corrected-to-normal vision.

Our investigations consisted of two studies (see below), both of which were carried out in the perception laboratory testing facility organized by the US National Cancer Institute at the annual meetings of the Radiological Society of North America. All radiologists participated with the proviso that they could only spare a few minutes of participation, which limited the number of test repeats, or ‘trials’, we could carry out on each subject.

Twelve practicing radiologists participated in Study 1. Six of the subjects were practicing mammography specialists with an average of 16.8 years (median, 15.5 years) of experience. The remaining six subjects were attending radiologists who specialized in another subspecialty of radiology and/or were trainees (residents or fellows) who had completed an average of 0.17 years (median, 0.5 years) of radiological experience.

Fifteen practicing radiologists participated in Study 2. Six of the subjects were practicing mammography specialists with an average of 12.8 years (median, 13.1 years) of experience. The remaining six subjects were attending radiologists who specialized in another subspecialty of radiology and/or were trainees (residents or fellows) who had completed an average of 6.4 years (median, 4.8 years) of radiological experience.

### 2.1. Study 1

The goal of this study was to characterize the effect of the AAA heuristic on the detection of cancer in mammograms by practicing radiologists. 

Stimuli. All images used in this study were radiologically vetted screening X-ray mammograms obtained from the Digital Database for Screening Mammography (DDSM) public repository [18]. We used mammograms that were professionally classified as negative for cancer (labeled ‘benign’ in the DDSM database) or positive for cancer (labeled ‘cancer’). Of the large number of cancer mammograms in the DDSM, we used those mammograms that contained a single, localized, and circumscribed microcalcification [18]. Equal numbers of positive and negative mammograms were chosen, so that the mammogram presented during any given trial had a 50% chance of being positive for cancer (Table 1).

Procedure. Previous studies [19,20] and our preliminary results [21] have shown that subjects, including clinicians, handle the probabilities more accurately when presented as ‘natural’ frequencies (e.g., percent chance) rather than fractions of 1. Therefore, all the probabilities in this study were expressed as percent chance. 

Prior to the actual data collection, subjects received detailed, illustrated instructions about the trial procedures. Subjects were encouraged to carry out practice trials before starting the actual trials to familiarize themselves with the procedure. The data from the practice trials were discarded.

Previous studies have shown that both the externally provided anchoring information and the subject’s own internal anchored positions have comparable biasing effects [14,15,22,23]. However, it should be noted that the anchoring information we manipulate in the present study is the externally provided anchoring information [2,4,22]).

Trial paradigm. During the actual data collection, each trial began when the subject indicated readiness by pressing a key on the computer’s keyboard; upon which, the subject was shown, for 2 s, an on-screen message stating the percent chance that the upcoming mammogram was positive for cancer (Figure 1A, left). We will refer to this estimate as ‘purported prior estimate ψ’. The subjects were told that this probability was determined by another radiologist, whereas, in actuality, this was a pseudorandom number generated de novo by a random number generator during each trial (also see below).

Subjects were then allowed unlimited time to provide an initial estimate of their perceived probability that the upcoming mammogram was positive for cancer (‘subject’s initial estimate α’) using an on-screen slider. A previously unseen mammogram was then presented for up to 0.2 s, 0.5 s, 1 s, 5 s, 10 s, and 60 s, depending on the trial (Figure 1A, middle), and the subjects were allowed to terminate the mammogram presentation and proceed to the next phase of the trial by pressing a designated key if they felt they had viewed the mammogram enough. The presentation of the mammogram was followed by a 0.5 s random dot mask. Subjects were then asked to estimate the probability that the mammogram they just viewed was positive for cancer (‘subject’s final estimate β’).

The trials were presented in a randomly interleaved fashion, so that each subject encountered each of the above six stimulus durations and two mammogram types exactly once. The experimental conditions were controlled and the data collected using custom-written scripts for the presentation software toolkit (neurobs.com) [24].

Rationale for using random numbers for purported prior probabilities ψ. Our experimental conditions were manifestly different from the actual conditions of the mammographic examination in many respects. Perhaps most importantly, the prior probabilities in our study were random numbers, whereas, in a clinical setting, this measure, if one is explicitly provided, would be an actual probability. Note, however, that using such radiologically vetted probabilities would have actually made it much harder to uncover the AAA effect to begin with, because, in such cases, these probabilities themselves would presumably be at near-perfect performance, thus presumably causing the initial and final estimates of the subjects to also track close to near-perfect performance. Thus, there would have been little or no room for adjustment and, therefore, little opportunity for the AAA effect to manifest itself, and we would have risked falsely concluding that there is no AAA effect in this case. Thus, using random nominal probabilities as we did, while admittedly clinically unrealistic, allowed the AAA effect, if any, to show through.

It is important to note that our IRB determined that our use of random numbers does not constitute deception under the applicable regulations and policies.

Data Analysis. Data were analyzed using scripts custom-written for the R (r-project.org) [24] and MATLAB (Mathworks.com) [25] software platforms. 

Receiver Operating Characteristic (ROC) analysis. Since the data from individual subjects (only 12 trials per subject) lacked sufficient statistical power, we carried out the ROC analyses using the data pooled across all subjects. This approach is subject to an important caveat that the actual decisions are made by each subject individually and not all subjects acting as a group. Nonetheless, such group-level ROC analyses have undeniable diagnostic usefulness (see, e.g., Reference [26]). Thus, in our case, the group-level ROC analyses tested the alternative hypothesis that the subjects’ pooled responses reflected the diagnostic information perceived by the subject sample.

We calculated the area under the ROC curve (AUC) using the R library [27]. 

Power analysis. We calculated the total number of trials needed (pooled across all subjects) to obtain a correlation between the purported prior estimate ψ and the subjects’ initial estimate α at correlation coefficient = 0.30, significance level = 0.05 (two-sided), and power level = 0.95. Significant correlation between ψ and α was chosen as the primary outcome of interest, because this meant that the prior estimate succeeded in producing a significant anchoring effect (see Results). This analysis (carried out using *R* library *pwr*) indicated that at least 138 trials were needed to achieve the requisite statistical power. The aforementioned 12 subjects performed 12 trials each, so that a total of 144 trials were carried out in this study.

### 2.2. Study 2

The goal of this study was to characterize the breast cancer detection performance of practicing radiologists with vs. without anchoring information. This study was identical to Study 1, except that the subjects performed the task with the anchoring information (conditions 1 and 2 in Table 2) or without (conditions 3 and 4 in Table 2).

## 3. Results

Study 1: Characterization of the effect of the AAA heuristic on detection of cancer in mammograms by practicing radiologists

In Study 1, practicing radiologists (*n* = 12) were shown a randomly generated number between 0 and 100 on a computer monitor. They were told this was the percent chance estimated by another radiologist that the upcoming mammogram contained cancer (‘purported prior estimate ψ’; Figure 1A, left panel, top). That is, this estimate represented the externally provided prior information in our case. The subjects were unaware this number was randomly generated.

Subjects then entered, using an onscreen slider, their own estimate that the upcoming mammogram contained cancer (‘subject’s initial estimate α’; Figure 1A, left panel, bottom). Following this, the subjects viewed the actual mammograms for various durations (0.2 s, 0.5 s, 1 s, 5 s, 10 s, or 60 s, depending on the trial; Figure 1A, middle panel); after which, they entered their estimate that the mammogram they had just viewed contained cancer (‘subject’s final estimate β’; Figure 1A, right panel).

The subjects’ initial estimates α were highly correlated with the purported prior estimates of the previous radiologist ψ (correlation coefficient *r* = 0.68, *df* = 142, *p* < 0.05; Figure 1B), indicating that the anchoring information (i.e., the purported prior estimate ψ) did succeed in producing a strong anchoring effect, as expected. That is, the subjects were strongly influenced by this ‘top-down’ information and tended to anchor their own initial estimates on this information. Moreover, the post hoc power analysis indicated that, at a statistical power of ≥0.95, this effect exceeded the requisite statistical power.

Note that, after viewing the mammogram, the subjects were required to estimate the chance that the mammogram they had just viewed contained cancer and that the sole relevant source of information for estimating this quantity was the mammogram itself. If the subjects solely relied on the mammogram information, their final estimates β would conform to the ground truth about the given mammogram (red and green dashed lines in Figure 1C; also, see Appendix A). However, the subjects’ final estimates of the cancer status of the mammograms substantially varied from the ground truth (i.e., were biased), regardless of whether the mammograms were positive or negative for cancer (red and green symbols in Figure 1C).

### Radiologists’ Final Estimates Are Adjusted Versions of Their Initial Estimates

To help characterize how the radiologists arrived at their final estimates, we plotted the size of adjustment δ*_i_* during a given trial *i* (i.e., the amount by which the subjects adjusted their final estimate β_i_ relative to their initial estimate α*_i_* during a given trial *i*; δ*_i_* = β*_i_* -α*_i_*) as a function of their initial estimate α*_i_* during that trial (Figure 1D). δ was highly anticorrelated with α, regardless of the cancer status θ of the mammogram (1-way ANCOVA; α: *F*(1,140) = 39.13, *p* < 4.5 × 10^−9^; θ: *F*(2,140) = 0.02, *p* = 0.89; also, see Figure 1D). This straightforwardly suggests that the reason why the final estimates were uncorrelated with the cancer status θ of the mammogram (Figure 1C) was because the radiologists arrived at their final estimates β by adjusting from their anchored positions α (Figure 1D), which were themselves highly correlated with the random ψ values (Figure 1B).

Post hoc modeling of the subjects’ final estimates confirmed that the actual cancer status of the mammogram indeed played a statistically insignificant role in the radiologists’ final estimates of cancer (Appendix A, row 2). Indeed, the only predictor that significantly accounted for the final estimates were the subjects’ initial estimates α (row 1). Neither the professional background of the radiologists (Appendix A, rows 4 and 5) or their demographic background factors such as age and gender (*p >* 0.05; not shown) played a significant role. The receiver operating characteristic (ROC) analysis indicated that the radiologists’ cancer detection performance was indistinguishable from random (Appendix A; area under the ROC curve (AUC) = 0.46; *p* > 0.05).

The result that the subjects performed at random levels is consistent with the fact that the anchoring information ψ that the subjects’ decisions were based on was itself random. This result is nonetheless surprising, because it suggests that trained radiologists can altogether ignore task-relevant empirical information in radiological images when they have access to anchoring information. One plausible explanation for this is that the subjects were under time pressure so that they were unable to scrutinize the mammograms sufficiently. Previous studies have shown that time pressure can induce subjects to resort to using heuristics [2,3]. However, our post hoc analyses indicated that the stimulus duration did not significantly contribute to the outcome (Appendix A, row 3). Moreover, subjects often took less than the allotted time before responding (see Appendix A; see the legend for details). Study 2 below revisits this issue.

Another conceivable explanation for the poor cancer detection performance of the radiologists is that they were, for whatever reason, unable to reliably detect cancer in the mammograms to begin with and therefore resorted to using the anchoring information. 

Study 2: Breast cancer detection performance of radiologists with vs. without anchoring information

We tested the above scenario in Study 2 using an independent sample of practicing radiologists (*n* = 15). This study was identical to Study 1, except in two respects (see Materials and Methods for details). First, depending on the trial, the subjects were allowed to view the mammogram for 0.2 s, 10 s, or for an unlimited duration. Second, the subjects performed the task with the anchoring information ψ (conditions 1 and 2) or without it (conditions 3 and 4; see Materials and Methods for details). 

Note that conditions 1 and 2 in this Study were identical to the corresponding conditions in Study 1. We hypothesized that if the subjects were unable to detect cancer in the mammograms to begin with, their performance should be at chance levels regardless of whether or not anchoring information was presented.

During the trials of Study 2 in which the radiologists received the anchoring information before they viewed the mammograms (conditions 1 and 2, in which the mammograms were negative and positive for cancer, respectively), their responses were comparable to those in Study 1 in all relevant aspects. In particular, the subjects’ cancer detection performance was indistinguishable from random (AUC = 0.48; *p* = 0.49 using randomization; Figure 2A,B and Table 3). Thus, the AAA effect from Study 1 was reproducible in Study 2 using an independent sample of practicing radiologists.

Conditions 3 and 4 in Study 2 (in which the mammograms were negative and positive for cancer, respectively) were identical to conditions 1 and 2, respectively, except that the purported prior estimate was blank (“- -”). In this case, the same radiologists were able to detect cancer in mammograms highly accurately (AUC, 0.71; *p* = 0.02; Figure 2C,D and Table 4). Thus, the chance level performance of the radiologists in conditions 1 and 2 was not attributable to an inability to accurately detect breast cancer in mammograms.

This straightforwardly indicates that the decision process of the radiologists was different depending on whether or not the anchoring information was available. In the former case, the subjects underweighted the visual evidence in mammograms in favor of the anchoring information. Viewed another way, the subjects used the AAA heuristic when they could and evidence-based decision-making when they had to.

Finally, post hoc modeling showed that the reaction time was not a significant contributing factor to the subjects’ final estimates in either study (Table 3 and Table 4, row 3). Similarly, when the modeling was repeated by stimulus duration for the reaction time, the results indicated that stimulus duration was not a significant contributor either (*p* > 0.05 for both studies; data not shown). Similarly, neither the subjects’ professional backgrounds (Table 3 and Table 4, rows 4 and 5) or demographic factors such as age and gender (*p* > 0.05 for both factors in both studies; data not shown) played a significant role. Thus, the subjects’ performances were not attributable to time pressure (q.v., References [2,3]).

## 4. Discussion

### 4.1. Interaction of Top-Down vs. Bottom-Up Factors during Diagnostic Decision-Making

Our results serve as the proof that the biasing effects of the AAA heuristic can, in principle, be so strong as to override the visual diagnostic evidence during mammographic decision-making. Our results also demonstrated that there are certain conditions, such as the availability of strong anchoring information in the present case, under which heuristic decision-making is the default mode and not the strategy of last resort when subjects make decisions under uncertainty. This finding is particularly important, because the resulting errors were large enough to reduce the subjects’ cancer detection performances to chance levels.

From a broader perspective, our results show that the biasing effects of AAA, previously demonstrated in the aggregate for subject groups evaluating verbal vignettes [2,4,28], persist in ‘retail’, case-by-case decision-making scenarios common in the real world, such as diagnostic decision-making in mammography. These results are of self-evident practical significance and suggest that, when clinicians are asked to provide a second opinion [29,30], it may be desirable to blind them to the first opinion.

From a more purely scientific perspective, our results characterize the interaction between the top-down heuristic influences and bottom-up empirical information that underlie many real-world decision-making scenarios and show that heuristics can override the sensory information. In this specific sense, the top-down effects are analogous to the effects of certain built-in priors that the brain uses in making decisions (e.g., the hollow face illusion) [31,32,33]. However, unlike such priors, the anchoring information is not built-in and can be ‘seeded’ externally, as the effects of ψ demonstrate in our case. 

### 4.2. Effects of the Anchoring Information on the Final Estimates

As noted above, previous studies have suggested that, in the case of AAA, subjects typically start from an anchored position and adjust their estimates until they are satisfied. Our results confirmed this notion and extended it to the context of mammography. They also confirmed and extended to mammography the previous studies that have shown that expertise training does not eliminate the use of heuristics and that highly trained experts can sometimes resort to using heuristics. Whether medical training has any effect on the radiologists’ use of the heuristics in general and the AAA heuristic remains to be seen.

Why does the anchoring information, when provided, degrade the cancer detection performance of the radiologists to random levels? There are two potential explanations for this. One possibility (‘scenario A’) is that the randomness of the performance reflects, and is caused by, the fact that the anchoring information is itself random. An alternate possibility (‘scenario B’) is that the anchoring information somehow more or less exactly counterbalances the sensory information in the mammograms, so that there is little or no net information that the subjects can act on.

Our results support scenario A and are inconsistent with scenario B. This is because our results show that the magnitude of adjustment is significantly anticorrelated with the anchored position (Figure 1D and Figure 2A). This means that the final estimate reflects the initial position, which, in turn, reflects the anchoring information provided to the subjects. That is, the subjects performed at random levels because they were operating based on random information, as envisioned in scenario A. Scenario B is contradicted by two lines of evidence. First, if scenario B applied to our case, then the subjects’ final estimates β would be expected to exactly contradict the ground truth about the cancer status of the mammogram, which was empirically not the case (q.v., Figure 1C and Appendix A). Second, if scenario B were true, then the subjects’ cancer detection performances would stay the same regardless of whether anchoring information was provided, because there was no change in the sensory information between the conditions. Yet, the performances drastically differed and did so based on the anchoring information. Together, these observations suggest that the reason the subjects performed randomly was that the information they relied on was random.

### 4.3. Important Caveats and Future Directions 

In addition to the various limitations of our study noted in context earlier, the following four caveats are especially worth noting within the limitations of this brief report. First, what our results demonstrated is that the AAA heuristic can override the sensory information under certain circumstances and not that the former always does override the latter. In this sense, our results illustrate a boundary condition and serve only as a proof-of-principle example and not necessarily a representative one. Indeed, it stands to reason that, under most conditions, the effects of the AAA heuristic are likely to be subtler and less pronounced than this. 

Second, note that, while our study used the established experimental procedures in this field, it did not, of necessity, simulate the actual clinical decision-making conditions. Note, however, that this is a limitation of all experimental studies of clinical decision-making and not a drawback specific to the present study. The usefulness of the present study is that it raises the possibility that prior information can have a strong, potentially undesirable influence on clinical judgments. This is consistent with a large number of previous studies that have demonstrated the effects of heuristics in other aspects of clinical decision-making. However, whether and to what extent such biasing effects do occur under real-world clinical conditions remain to determined.

A related concern is that our results are attributable to the relative abundance of mammograms positive for cancer in our case (50%) compared to the actual incidence of breast cancer in screening mammograms (<0.005% [34,35]). If this were the case, one would expect to see a disproportionate number of false negatives (i.e., misses) in our case. This was the case in our study (Pearson’s chi-square test, *p* > 0.05). This outcome can be visually discerned from the fact that the red symbols are roughly uniformly distributed along the *y*-axis of Figure 1C and Appendix A.

A third major caveat is whether the radiologists would recognize upon more prolonged testing that the purported prior estimate ψ was indeed random or at least unreliable. The present preliminary study was unable to address this important issue, because each radiologist in our study was able to perform only 12 trials (see above). In this narrow sense, the conditions of our study were somewhat similar to clinical contexts, where the given radiologist may have access to a previous radiologist’s judgement in only a few cases and not for a long series of cases in a row. A related issue is the extent to which the radiologists’ implicit trust in the purported prior estimate made them attach greater weight on the prior estimate than they otherwise would have (also, see below). For instance, it stands to reason that the radiologists would have more or less ignored the prior estimates ψ if they had been told that the estimates were from novice nonprofessionals.

Finally, while our study used mammography as an illustrative example to help highlight the potential power of the heuristic, it remains an open question as to whether and to what extent our results are generalizable in other contexts, such as other diagnostic tasks, other stimuli, differences in subject training and expertise, the use of computer-assisted detection/diagnosis (CAD) technologies, etc. A related issue is the extent to which other known heuristics play a comparable role in diagnostic decision-making, which our study did not address. Further studies are needed to address these important issues raised by our study.

## Figures and Tables

**Figure 1 diagnostics-12-00105-f001:**
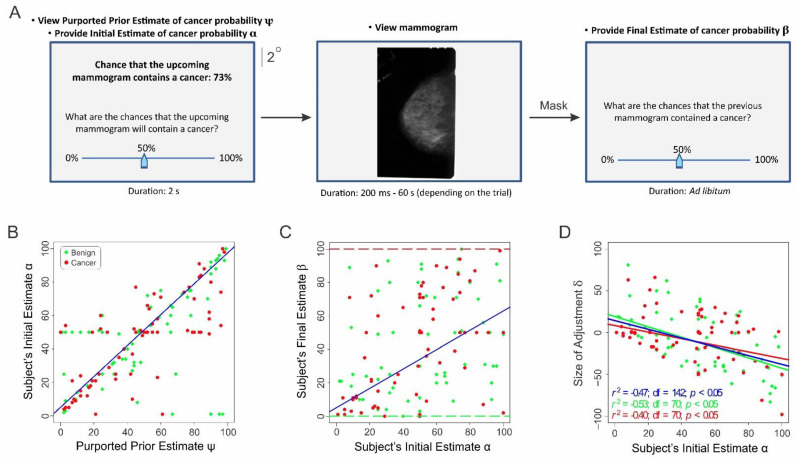
Task paradigm and results from Study 1. (**A**) Task paradigm. Each trial started with the onset of a central fixation spot, and each mammogram was followed by a 0.5 s random dot mask (not shown). Not drawn to exact scale. (**B**–**D**) Results. (**B**) Subjects’ initial estimate α as a function of the previous radiologist’s estimate ψ. (**C**) Subjects’ final estimate β as a function of their initial estimate α. The red and green dashed lines denote the expected responses for mammograms positive and negative for cancer, respectively. (**D**) The amount of the subjects’ adjustment δ as a function of their initial estimate α. The red, green, and blue dashed lines denote the best-fitting regression lines for all the mammograms, mammograms positive for cancer, and mammograms negative for cancer, respectively.

**Figure 2 diagnostics-12-00105-f002:**
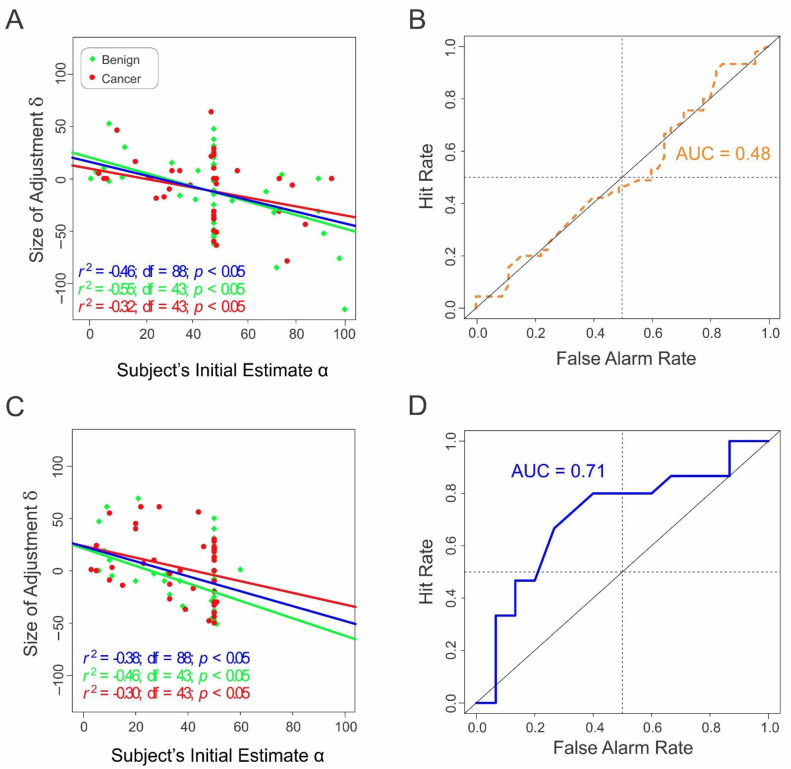
Results from Study 2. (**A**) Radiologists’ final estimates as a function of their initial estimates in the presence of anchoring information. (**B**) ROC analysis of the radiologists’ final estimates in the presence of anchoring information. (**C**) Radiologists’ final estimates as a function of their initial estimates in the absence of anchoring information. (**D**) ROC analysis of the radiologists’ final estimates in the absence of anchoring information.

**Table 1 diagnostics-12-00105-t001:** Experimental conditions in Study 1.

Condition #	Anchoring Information	Cancer Status of the Mammogram
1	Provided	Negative
2	Provided	Positive

Note that, in this study, anchoring information was provided to the subjects in both experimental conditions.

**Table 2 diagnostics-12-00105-t002:** Experimental conditions in Study 2.

Condition #	Anchoring Information	Cancer Status of the Mammogram
1	Provided	Negative
2	Provided	Positive
3	Not provided	Negative
4	Not provided	Positive

Note that the first two conditions were identical to the two conditions in Study 1 above, so that half of this study aimed to replicate Study 1 in an independent set of subjects. A total of 180 trials (15 subjects × 12 trials each) were carried out in this study.

**Table 3 diagnostics-12-00105-t003:** Contribution of the various explanatory variables to the final estimates γ when anchoring information was available in Study 2 (conditions 1 and 2).

Row #	Explanatory Variable	Estimated Coefficient	Standard Error	*t* Value	*p*-Value
1	Subjects’ initial estimate α	0.48	0.11	4.40	3.15 × 10^−5^
2	Cancer status of the mammogram (−ve vs. +ve for cancer) θ	0.54	4.63	0.12	0.91
3	Reaction time *r*	1.94 × 10^−4^	1.87 × 10^−4^	1.04	0.30
4	Radiological specialty of the subject	10.56	14.36	0.74	0.46
5	Length of radiological experience of the subject	−0.38	0.22	−1.72	0.09

**Table 4 diagnostics-12-00105-t004:** Contribution of the various explanatory variables to the final estimates γ when anchoring information was unavailable in Study 2 (conditions 3 and 4).

Row #	Explanatory Variable	Estimated Coefficient	Standard Error	*t* Value	*p*-Value
1	Subjects’ initial estimate α	0.34	0.19	1.83	0.07
2	Cancer status of the mammogram (−ve vs. +ve for cancer) θ	9.16	5.77	1.59	0.12
3	Reaction time *r*	−7.14 × 10^−4^	3.37 × 10^−4^	−2.12	0.04
4	Radiological specialty of the subject	5.69	18.08	0.32	0.75
5	Length of radiological experience of the subject	−0.28	0.29	−0.98	0.33

## Data Availability

The data presented in this study are available in a de-identified form from the corresponding author upon reasonable request. The data are not publicly available owing to applicable privacy guidelines.

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
