# Peer review of "Biasing Influence of ‘Mental Shortcuts’ on Diagnostic Decision-Making: Radiologists Can Overlook Breast Cancer in Mammograms When Prior Diagnostic Information Is Available"

_diagnostics, 2022, doi:10.3390/diagnostics12010105_

Round 1
Reviewer 1 Report
Very interesting project! Would be interesting to see how percentage chance of cancer offered by AI software affect radiologist interpretation.
Author Response
> Very interesting project!
We thank the Reviewer for this kind comment.
> Would be interesting to see how percentage chance of cancer > offered by AI software affect radiologist interpretation.
We fully concur. We have tweaked the relevant paragraph in the Discussion section to include this idea.
Reviewer 2 Report
Nice work, Good job.
Interesting to see in a prospective study and real clinical scenarios.
Author Response
> Nice work, Good job.
We thank the Reviewer for this kind comment.
>Interesting to see in a prospective study and real clinical scenarios.
We fully concur. As the reviewer no doubt noticed, the original manuscript mentioned the idea of prospective studies in the 3rd paragraph of the Discussion section. In the revised manuscript, we have revised the relevant section of the Discussion section to more explicitly mention that further studies are needed to determine if/how our results generalize to real-world clinical conditions.
Reviewer 3 Report
I congratulate the authors for this novel and valuable work. There are several minor comments that can improve the manuscript:
- Introduction: please add a sentence regarding the main types of heuristics: "for example, representativeness, anchoring and adjustment, and availability."
- Introduction: An anchoring and adjustment (AAA) decision can also be applied in scientific reasoning. A good example of that is the article by Akbari et al. (https://doi.org/10.1016/j.mito.2021.12.001), in which the authors anchored the mitochondria as the determinant of response to immunotherapy and provided evidence-based adjustments to prove their hypothesis. It is suggested authors mention this issue.
- Line 77-81: "Six of the subjects were practicing mammography specialists with an average of 16.8 years of experience. The remaining six subjects were attending radiologists who specialized in another subspecialty of radiology, and/or were trainees (residents or fellows) who had completed an average of 0.17 years of radiological experience". Due to the small number of cases per group, it is suggested to use median and range for reporting the distribution of participants' experiences.
- This comment also includes the paragraph between lines 82-86.
- Methods: Please completely describe the objective(s) of studies 1 and 2. And omit the description of studies 1 and 2 from the subheading. I mean (for example) the subheading of "Study 1: Characterization of the effect of the AAA heuristic on detection of cancer in 87 mammograms by practicing radiologists" can be changed into "study 1".
- Table 1 and 2. It is suggested to add demographic variables (e.g., age and sex)
Author Response
> I congratulate the authors for this novel and valuable work. There are several minor comments that can improve the manuscript:
We greatly appreciate this comment.
> 1. Introduction: please add a sentence regarding the main types of heuristics: "for example, representativeness, anchoring and adjustment, and availability."
We fully concur. We have complied.
> 2. Introduction: An anchoring and adjustment (AAA) decision can also be applied in scientific reasoning. A good example of that is the article by Akbari et al. (https://doi.org/10.1016/j.mito.2021.12.001), in which the authors anchored the mitochondria as the determinant of response to immunotherapy and provided evidence-based adjustments to prove their hypothesis. It is suggested authors mention this issue.
We thank the Reviewer this suggestion. We have revised the relevant portion of Introduction to mention the idea that AAA effects occur in scientific research, too. We have cited three references to the work of Kahneman and Tversky on this topic.
> 3. Line 77-81: "Six of the subjects were practicing mammography specialists with an average of 16.8 years of experience. The remaining six subjects were attending radiologists who specialized in another subspecialty of radiology, and/or were trainees (residents or fellows) who had completed an average of 0.17 years of radiological experience". Due to the small number of cases per group, it is suggested to use median and range for reporting the distribution of participants' experiences.
We appreciate the suggestion. We have added the median info to the revised version. (We decided to retain the info about the means as well, so that the readers can get some idea of the skewness of the samples.
> 4. This comment also includes the paragraph between lines 82-86.
Thank you very much, we have modified them.
> 5. Methods: Please completely describe the objective(s) of studies 1 and 2. And omit the description of studies 1 and 2 from the subheading. I mean (for example) the subheading of "Study 1: Characterization of the effect of the AAA heuristic on detection of cancer in 87 mammograms by practicing radiologists" can be changed into "study 1".
We fully concur with this sentiment, and have indeed modified them accordingly.
> 6. Table 1 and 2. It is suggested to add demographic variables (e.g., age and sex)
We appreciate where the Reviewer is coming from. We have noted in the revised manuscript that neither of these demographic variables make statistically significant difference.
However, we decided against actually adding this Tables 1 and 2, for the following reason: First, the models shown in these Tables in the original version were determined using standard model selection procedures, so that these are the most parsimonious/most optimal models that best account for the observed data. During these model selection procedures, the aforementioned demographic variables were not retained (i.e., only the variables shown in Tables 1 and 2 were retained). Therefore, adding the demographic variables will amount to the well-known artifact of overfitting.
Therefore, we felt that the approach we have adopted (whereby we show the most parsimonious models in the Tables, but mention in the text that age and gender don’t make a significant difference) satisfies both imperatives and optimally balances them. We hope that the Reviewer appreciates that is a principled approach.